# Socio-economic urban scaling properties: Influence of regional geographic heterogeneities in Brazil

**Caio Porto de Castro**[1☯]*, **Gervásio Ferreira dos Santos**[1,4☯], **Anderson Dias de Freitas**[1☯], **Maria Isabel dos Santos**[1☯], **Roberto Fernandes Silva Andrade**[1,3☯], **Maurício Lima Barreto**[1,2☯]

**1** Centre for Data and Knowledge Integration for Health, Instituto Gonçalo Muniz, Fundação Oswaldo Cruz, Salvador, Brazil, **2** Institute of Collective Health, Federal University of Bahia, Salvador, Brazil, **3** Institute of Physics, Federal University of Bahia, Salvador, Brazil, **4** Economics Faculty, Federal University of Bahia, Salvador, Brazil

☯ These authors contributed equally to this work.
* caioporto@ufba.br

**Data Availability Statement:** The data used in this article can be found at https://doi.org/10.17605/

## Abstract

The recent efforts dedicated to understanding important features and consequences of city growth have profited from the scaling approach to urban indicators. This kind of analysis can be conveniently used to investigate the impact of geo-economic transformations, like fast urbanization and industrial development, which occurred in continental size countries (e.g., India, China, and Brazil) during the past half-century. Profiting from high quality data, this work explores how scaling relationships among urban indicators are influenced by strong regional heterogeneities in Brazil. It is based on economic, infrastructure and violence related data sets for the time interval 2002-2016. Results indicate that regional specificities related to infrastructure, economic development, and geography have a larger influence on the absolute value of the urban indexes. Regional scaling similarities and differences among Brazilian regions were also uncovered. Interesting enough, the results indicate that the richest and poorest Brazilian regions share similar scaling behavior, despite all huge different local influences. By contrast, the results for the two richest regions, with similar average values of urban indexes and the same kind of local influences have rather different scaling properties. Thus, scaling analysis suggests that distinct political solutions might be necessary to improve life's quality, even for two regions with similar average values of urban indicators.

## Introduction

Most of the world population already lives in urban areas, and proportion will most probably increase in the next decades (https://data.worldbank.org/indicator/sp.urb.totl.in.zs). Cities scale over a huge range of population size, from just a few thousand to tens of millions of inhabitants, creating many facets as infrastructural, economic, social and spatial complex

OSF.IO/K342Q which was extracted from the references 32 and 33.

**Funding:** This work was supported by the Brazilian agency CNPq through grant No. 305060/2015-5 (R.F.S.A.), by the Wellcome Trust (UK) through the project Salud Urbana en América Latina (SALURBAL) (C.P.C., G.F.S., A.D.F., M.I.S.). R.F.S. A. acknowledges the support of the National Institute of Science and Technology for Complex Systems (INCT-SC Brazil). The funders had no role in study design, data collection and analysis, decision to publish, or preparation of the manuscript.

**Competing interests:** The authors have declared that no competing interests exist.

systems. Living in cities has proportioned great advantages for economy, creativity and, because of the given facilities to access assistance services and programs, also improve population health. On the other hand, urbanization also brings potential risk to health with problems associated with high population density, such as criminality, landscape changes and complex social interactions of many individuals.

In the last decade, seminal works by West, Bettencourt and other authors have presented and motivated several empirical and theoretical description of urban scaling [1–9]. This perspective does not immediately provide a causal explanation among urban indicators and population size, but rather identifies general and systematic relationship properties among them. Causality assessments require the understanding of how individuals interact with each other, which is still unclear and object of investigation. The scaling approach, which applies to broad range of urban indicators [1–19], has suggested that the population size of the cities is a main determinant of the intensity of socio-economic activity in urban areas.

The allometric scaling relations indicate that rates of human social behavior, urban infrastructure properties, patterns of consumption, diseases, innovation, have worldwide prevalence [1–20]. Thus, regardless their specific historical and cultural backgrounds, on the average, the cities are non-linear scaled versions of each other. In fact, such non-linearities are reality in many agglomeration phenomena, once they are results of the complex interactions in social dynamics and cities organization [3, 4, 19, 21–25].

Despite the recent advances in urban scaling analyzes of socioeconomic activities and health conditions, additional efforts are still necessary to include other social dimensions, like the effects of evident regional disparities in the country. Such regionalities are mainly present in continental size countries as China, India, USA, Russia and Brazil [26, 27]. However, conclusive studies uncovering regional dependence of scaling analyzes for most of these countries are still missing. Based on these considerations, this paper discusses scaling analyzes carried out for different socioeconomic and health dimensions of the Brazilian geographical regions, a country internationally known by its huge inequality levels of income, social and health conditions, as well as regional disparities.

The geography of inequality of regional income and population for the five macro regions of Brazil is illustrated in the Fig 1. As can be seen in the Fig 1(a), the Southeast region historically concentrates more than 50% of the national Gross Domestic Product (GDP). A very slow declining behavior of this share has been observed in the last decades, when Midwest and North regions consistently captured parcels of the GDP. In the same way, Fig 1(b) shows the time evolution of population distribution. The poorest Northwest region concentrates almost 1/3 of population. The long time persistence of such regional inequality pattern actually brings serious socioeconomic consequences for the country's development, as it affects income, health, violence, innovation, productivity and other key performance indicators. The evolution of the regional urbanization process in Brazil is shown in Fig 2. As can be verified in the Fig 2(a), the process is convergent among the regions. In the highest and lowest urbanized regions, respectively the Southeast and Northeast, more than 90% and 70% of population already lived in the urban areas in 2010. The same process of convergence is verified in the concentration of population in cities with more than 100M inhabitants. The convergence of urbanization process and population living in large cities rise important questions regarding scaling behavior in different Brazilian regions. Nevertheless, it is important to note that the Midwest and North regions have the widest territorial parcels of the country (see Fig 3) and, at the same time, the lowest populational densities. The work addresses a further important aspect, namely the meaningful comparison of the actual values of a given urban index among the ensemble of analyzed cities. Ranks that consider the simple per capita rates can be significantly distorted when the

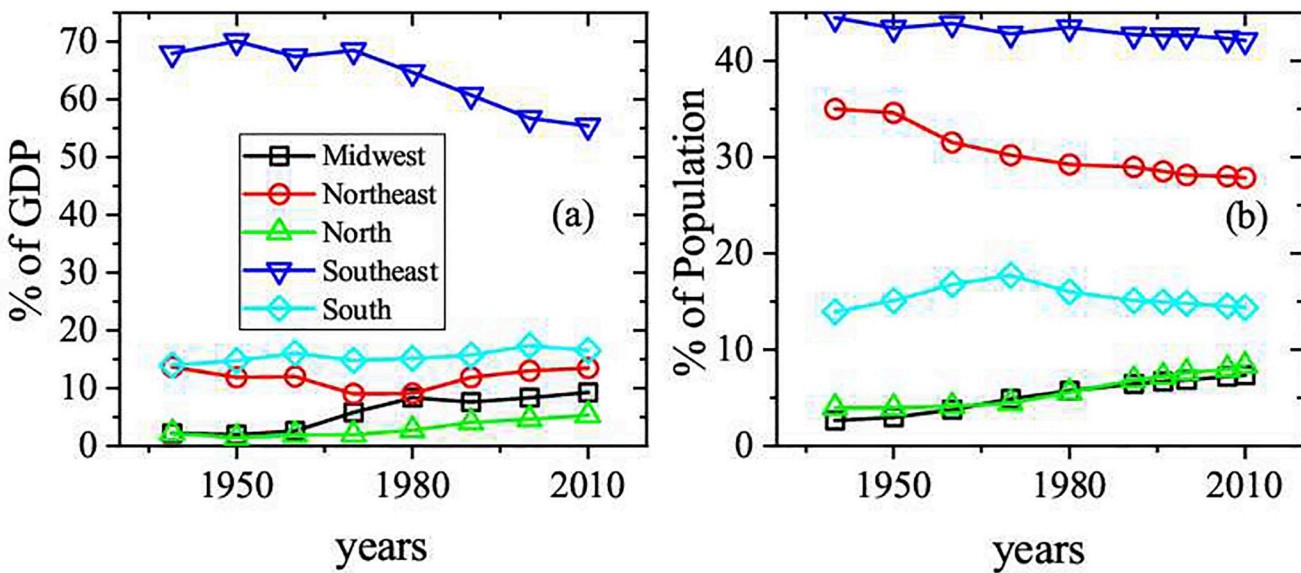

**Fig 1. Share of (a) GDP and (b) population of the Brazilian macro regions, 1939-2010.** Source: Brazilian Institute of Geography and Statistics and Brazilian Institute of Applied Economic Research.

considered index shows nonlinear dependence with respect to the population size. Therefore, in such cases it may be more convenient to use ranks based on the normalized residuals with respect to the expected results.

In the following sections we present the scaling analysis of an economic, violence and health, infrastructure and education urban variables, considering the five Brazilian political regions: South, North, Southeast, Northeast and Midwest. We have verified that regions with

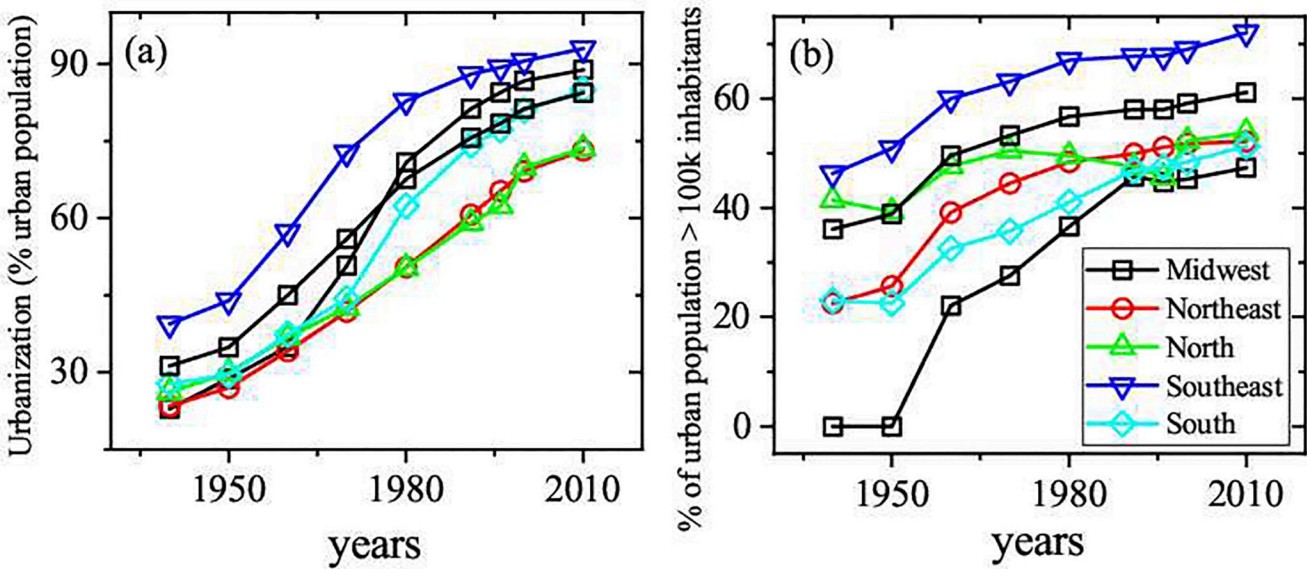

**Fig 2. (a) Urbanization and (b) urban population in large cities in the Brazilian macro regions, 1939-2010.** Source: Brazilian Institute of Geography and Statistics and Brazilian Institute of Applied Economic Research.

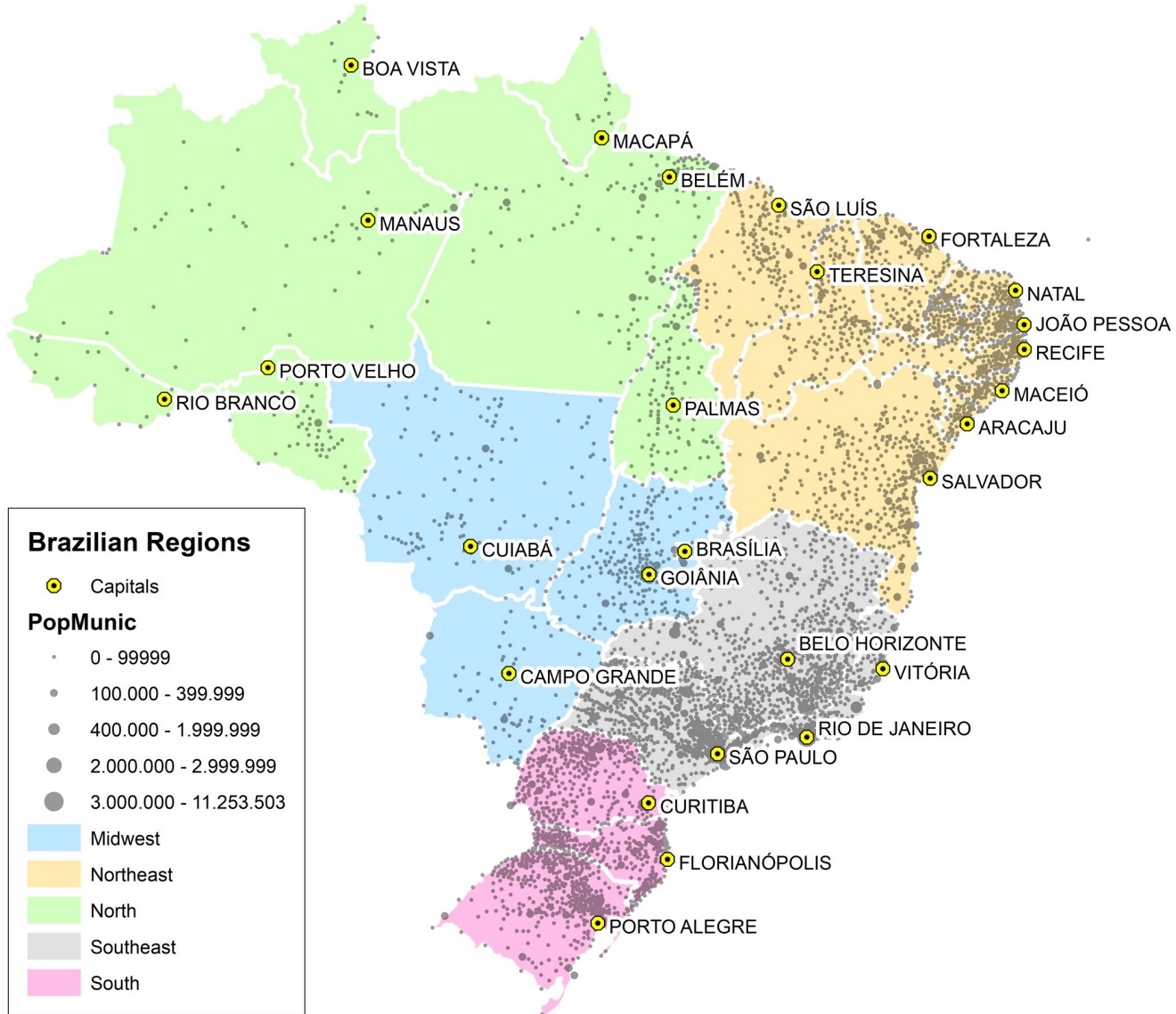

**Fig 3. Here we show how the Brazilian cities are spatially distributed over the country.** The capital of each federal state are written and the Brazilian regions are also characterized by different colors. Map generated with the free and open source geographic information system (QGIS) version 3.4 LTR.

similar economic development, infrastructure, and level of industrialization, such as South and Southeast do not, necessarily, share analogous scaling features. On the other hand, we have identified similar scaling dependence between one of the poorest and richest regions, suggesting that despite difference of the absolute value between the urban indicator of these regions, the biggest cities are equally different of the smallest cities of their respective regions. Such findings are important once it may indicate that general political interventions in continental countries may not be adequate. Finally, we emphasize the importance of comparing cities using scaling analysis in completeness with per capita indexes, once that most of the urban indicators have presented a non-linear dependence with population size. As we show, ranking cities by per capita indexes may not be similar to scaling indexes leading to either sub or over estimation of the urban variable effect.

## Methods

The usual per capita indicators assume that a given urban indicator, $Y$, depends linearly with population size, $N$, as $Y \propto N$, masking the non-linear interactions in the social dynamics of the cities. However, several relationships between system size and pertinent indicators in urban systems, both related to social [28] and infrastructure [29] aspects, have shown a non-linear behavior described, on average, by a power-law relation,

$$Y = Y_0 N^{\beta}, \tag{1}$$

where $Y_0$ is a normalization constant and $\beta$ is the scaling exponent, which reflects intrinsic properties of $Y$. Similar results have been obtained in the realm of complex adaptive biological systems [30]. Given a per capita variable written by $y_{per} = Y/N$, one may conclude from Eq 1 that the variation of $y_{per}$ satisfies,

$$\frac{dy_{per}}{dN} = (\beta - 1)Y_0 N^{\beta-2}, \tag{2}$$

so that the fraction variation of the population size reads

$$\frac{\Delta y_{per}}{y_{per}} \approx (\beta - 1)\frac{\Delta N}{N}. \tag{3}$$

Consequently, $\beta = 1$ leads to $\Delta y_{per} = 0$, which means that $Y$ is linearly dependent on $N$. In this case, comparing cities by their per capita rates constitutes an adequate approach. On the other hand, assuming $\beta > 1$ leads to $\Delta y_{per} > 0$, indicating that the variable $Y$ increases faster than the population size. Finally, for $\beta < 1$, one has $\Delta y_{per} < 0$, which means that the variable $Y$ increases slower than the population size, corresponding to an economy of scale. Thus, the allometric analysis hints that superlinear and sublinear beahavior are not random events, but suffer the influence of other features related to agglomeration size. Curiously, many such indexes have been classified similarly for different sets of studied cities worldwide [9, 13].

A qualitative assessment of the $\beta$ values for urban indicators depends on their intrinsic nature. This means that, both $\beta > 1$ and $\beta < 1$ may indicate either a positive or negative (desired or undesired) aspect, depending on which urban indicator is being analyzed. For example, while a good scenario is indicated by superlinear behavior of urban indicators like wages, income, gross domestic product, we can not say the same for pollution rates, crimes and disease incidence. On the other hand, cases of sublinear behavior of urban indicators related to infrastructure and material (length of electrical cables, gas station) represent an optimized scenario. In general, Eq 1 is transformed logarithmically as

$$\log(Y) = \log(Y_0) + \beta \log(N). \tag{4}$$

Working with the logarithm (on base 10) of the original data, we have used the Ordinary Least Squares regression (OLS) to estimate the log-linear relation (Eq 4), where the slope provides an estimation for $\beta$.

The average behavior of the urban factors is expressed by Eq 1. However, to measure the statistical fluctuation over such average behavior and, consequently, to access the local characteristic of every individual city (index $i$), we have estimated the urban indicator residuals (UIR) $\zeta_i$ [31],

$$\zeta_i = \log\frac{Y_i}{Y} = \log\frac{Y_i}{Y_0 N^{\beta}} \tag{5}$$

where $Y_i$ is the value of the urban factor for a specific city. Then, cities will be compared by

their $\zeta_i$ values, which provide a quantitative estimation of development or under-development with respect to the average behavior.

## Discussion and results

The Brazilian data at city level is quite consistent, and most urban indexes cover all Brazilian cities. In order to show how scaling analysis has the potential to uncover regional discrepancies, we have studied 4 socioeconomic variables: the Gross Domestic Product (GDP), the number of homicides, the length (in kilometer) of the water distribution system (WDS), and the number of high knowledge employees in the service sector (SHT). These variables are able to capture essential aspects related to economic, violence, infrastructure and technology standards of the cities. Of course, other choices are also possible. The analyzed data was collected from the department of computing of the Brazilian health care system DATASUS [32] and from the Brazilian Institute of Geography and Statistics (IBGE) [33] from 2002 to 2016 for near 5000 Brazilian cities. From DATASUS, where a set of health information of all Brazilian city is stored, we have extracted the total of homicides of every Brazilian city aggregated by year. The IBGE is a official Brazilian agency responsible for collecting and estimating social-economic information. Based on data from DATASUS and IBGE many public policies are planned at all government levels. We have analized 6 different sets of cities: the first one comprises all Brazilian cities, while the other are subsets of cities in the 5 Brazilian geographical regions (North, Northeast, Midwest, Southeast and South). The distribution of cities among the different regions in Brazil is the following: approximately 32% in the Northeast, 30% in the Southeast, 20% in the South and 8% in the North and Midwest regions. As shown in Fig 3, most of the cities are on the coast and while, as compared with other regions, the Midwest and North have lower city density and concentrate less population. The percentage of population by region is show in Fig 1b. The population in the cities range from $\sim 10^3$ to $1.2 \times 10^7$ (São Paulo). This work is divided into two sequences of investigation: in first place, we evaluated the scaling dependency of the urban indicators with respect to the population size. After that, we proceeded with a local characterization of cities by the relative parameter $\zeta_i$, which quantifies the deviation of the city indicators from the average scaling expected behavior.

From the Ordinary Least Squares regression regression, we have initially estimated the baseline average behavior of the urban indicators. For the purpose of obtaining more information on the influence of the scatterd data on the linear regression, we also provide results from by Nadaraya-Watson Kernel (NWK) regression. Fig 4 shows the log-log plots of the urban indicators with respect to the population size. The almost general good consonance between NWK regression and the linear regression supports the assumption that the dependency of GDP, homicides, water distribution system and the number of high knowledge employees in the service sector with respect to population size may be expressed by Eq 4, indicating a power-law scaling dependency. The main discrepancy revealed by NWK refers to the large scale regime in the midwest region.

Based on the evidences of the power-law scaling dependency, we have studied the longitudinal behavior of $\beta$'s values, related to every Brazilian region and all Brazilian cities (see Fig 5). For a solid interpretation, it is important mentioning the Brazilian economic heterogeneity between and within the Brazilian regions, as we have shown in the supporting information. In addition, we have also calculated the degree of Spatial stratified heterogeneity (SSH) [34] for the four socioeconomic variables, which is expressed by the parameter $q = 1 - \frac{1}{N\sigma^2} \sum_{i=1}^{L} N_i \sigma_i^2$. Considering Brazil as our sample, $L = 5$ is the number of strata in correspondence with the number of Brazilian regions. The number of units (cities) is represented by $N$, and $\sigma^2$ is the variance of the socioeconomic variable of all Brazilian cities. $N_i$ and $\sigma_i^2$ are the number of cities

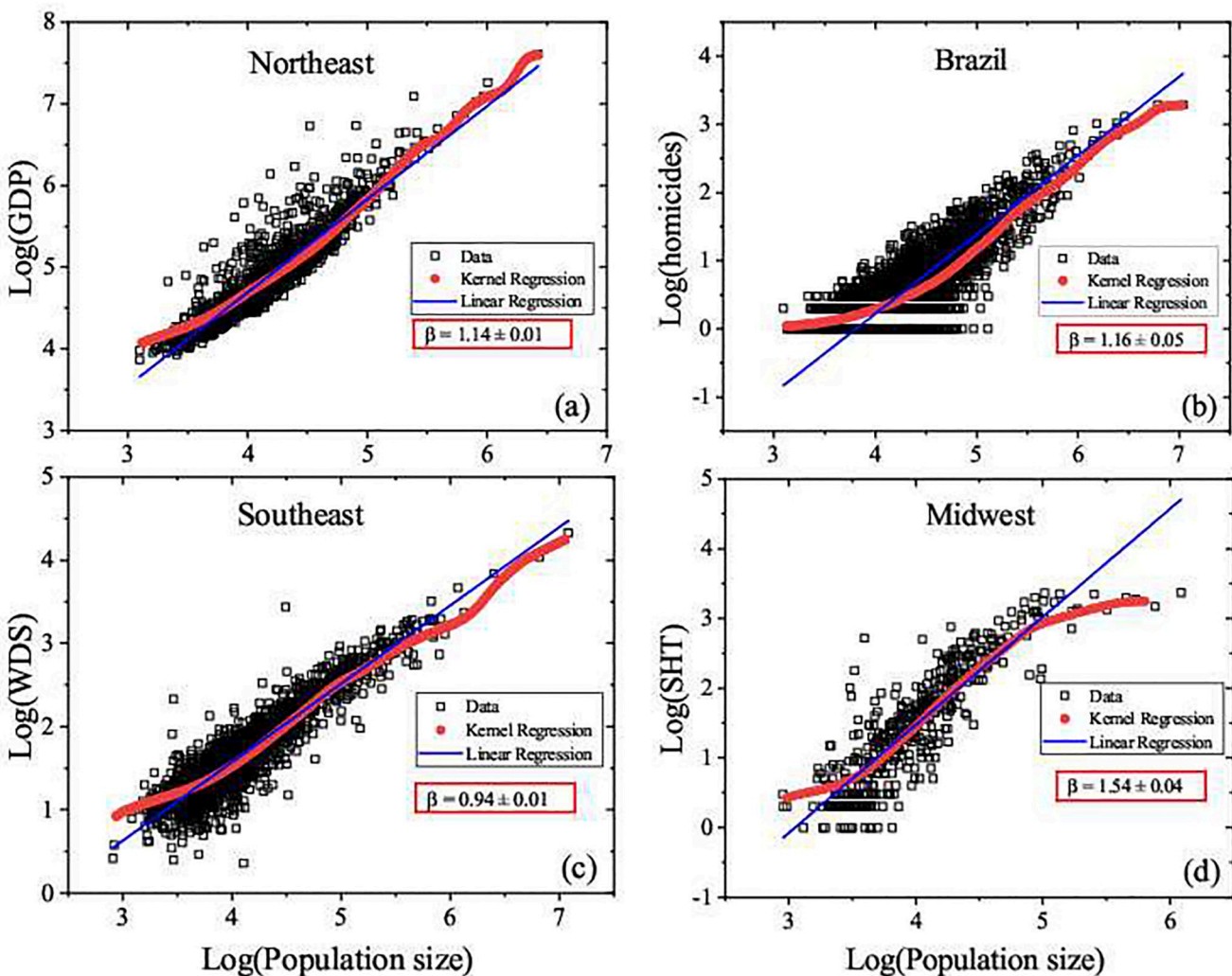

**Fig 4. Examples of typical urban indicator × population size plots in double logarithm scale in base 10.** (a) 2010 Brazilian Gross Domestic Product, GDP, of the ensemble Northeast, (b) 2007 homicides of the ensemble Brazil, (c) 2016 water distribution system, in Kilometers, of the ensemble Brazil and (d) 2006 numbers of high knowledge employees of the Southeast ensemble. The red and blue lines correspond to the NWK regression and linear regressions, respectively. The NWK results show that despite the fluctuations the urban indicators may be described by Eq 1. In the red boxes are indicated the values $\beta$, corresponding to the slope of the blue line described by Eq 4. For the $\beta$ values of all ensembles for the whole time range see Fig 5.

and the variance of the socioeconomic variable of cities for a specific Brazilian region. As the definition of $q$ is based on the ratio between the variance within strata and the variance of the entire study area, the interpretation of the limiting values of $q \in [0, 1]$ is the following: (a) as the values of $q$ become closer to 1, the heterogeneities within strata are more similar than those between strata; (b) as the values of $q$ become closer to 0, the heterogeneities between strata are more similar than those within strata. Therefore, the results in Table 1, where q-values are closer to 0 than 1, can be interpreted as heterogeneities within regions are higher than between them. The larger values of $q$ for WDS and SHT can express the fact that discrepancies in education, and infrastructure between the Brazilian regions are relatively larger than those in wealth and violence. It is well known that the South and Southeast regions are the most rich and developed Brazilian regions. Their GDP average value present an immense gap compared with those of other Brazilian regions. Also, the South and Southeast regions present larger

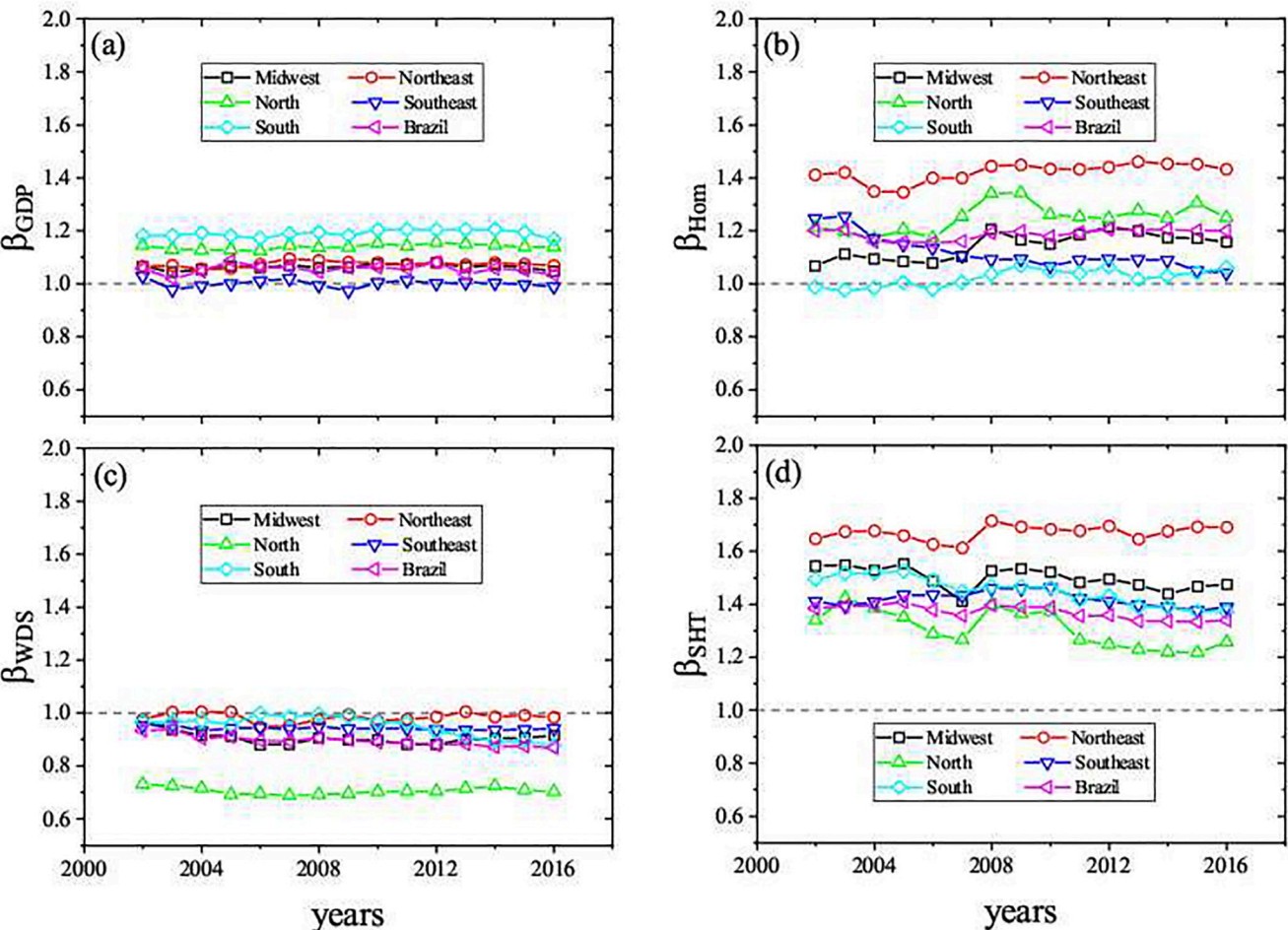

**Fig 5. Longitudinal evaluation of $\beta$ for all ensembles from 2002 to 2016: (a) GDP, (b) homicides, (c) water distribution system, and (d) high knowledge employees.** For a better visualization of the plots, we have hidden the error bars which are approximately 0.01 for the $\beta_{GDP}$, 0.1 for $\beta_{Hom}$, [0.07, 0.15] for $\beta_{WDS}$, and [0.05, 0.1] for $\beta_{SHT}$.

consumer market, better connectivity among cities, general infrastructure, population density as well as they have almost 50% of the total number of Brazilian cities.

A huge difference exists between the Southeast and Northeast regions if they are compared by their absolute GDP values, as they are, respectively, the richest and poorest regions. Despite that, we have found that the GDP of Southeast and Northeast regions share a similar intensity

**Table 1. $q$-values obtained from the 2016 data indicating that all Brazilian regions present high similar heterogeneity within them.**

| Socioeconomic variable | Geodetector $q$ |
|---|---|
| GDP | 0.00238 |
| Homicide | 0.00304 |
| WDS | 0.11289 |
| SHT | 0.03603 |

Large concentration of wealth, infrastructure, high level education employees and violence within the regions, which are mainly observed in the metropolitan area of every Brazilian state, are responsible for the values of $q$ close to zero.

scaling relation, characterized by a strong superlinear dependence. In Table 1, the small values of $q$ for the socioeconomic variables indicates that heterogeneity within the regions are more relevant than those between them. On the other hand, the largest value of $q$ associated with WDS indicates a comparatively smaller infrastructure discrepancy between the richest regions with the poorest ones. This can be explained, among other reasons as political and historical, by the fact that the poorer regions have globally had less investments in infrastructure. In contrast to that, the South and Midwest regions have much weaker superlinear dependency. These results make evident that, despite the fact that small and medium cities of the Southeast are more developed than their counterparts in the Northeast, the comparison of these subsets with those formed by larger cities, in their respective regions, indicates that, on the average, the small cities suffer a similar kind of local inequality with respect to wealth distribution. A comparison among the South, North, and Midwest regions shows a small scaling dependency, irrespective of the global GDP disparity among them.

In an ideal scenario, higher GDP values would indicate how intense the economic activity of a city is, since GDP is an indicator of consumption, a good proxy of welfare. The comparison among small, medium and large cities within either the Midwest or South regions shows that they have quite similar proportional growth of offer of public services. On the other hand, analyzing Southeast or Northeast regions, the difference between small, medium and large cities would be more evident, once the GDP grows much faster than the population sizes. The GDP is just one of several socioeconomic quantities that have shown superlinear behavior. Results for the complete set of Brazilian cities show that the GDP presents a low scaling dependency indicating that, despite disparities in inter-regional wealth distribution and intra-regional scaling properties, small and medium cities have strong contribution to the Brazilian GDP. This is a clear indication that, for large countries with strong regional differences, the scaling analysis based only on national data is not able to reveal huge non linear discrepancies existing at a regional scale. It is also noteworthy that these indicators have been quite stable for the whole period of analysis.

One may also conjecture whether such scenario would be more perceptible in urban indicators of violence, where the individual safety feeling is related not only to the number of cases, but also depends on the population size. Results for $\beta_{Hom}$ along the considered 15 years indicate that almost all regions present stronger dependency as compared with $\beta_{GDP}$, with exception for the South and Southeast regions. The strong scaling dependency expresses the fact that violence in large cities is more noticeable than small cities. The fact that the Southeast region has shown a decreasing scaling dependency in more recent years should not be interpreted by reduction of homicides cases. Actually, the number of homicides in Brazil has grown in most cities. The interpretation based on city sizes indicates that, in the South, North, and Northeast regions, the number of homicides has increased rather proportionally in all city scales, whereas in the Midwest larger cities have become more violent. The same happens with the small and medium cities in the Southeast region.

The water distribution system, an infrastructure variable, presented sublinear scaling dependency. It is possible to identify in Fig 5(c) the strongest sublinear dependency for North region. It should be noted that this Brazilian region is by far the least optimized one in terms of urban structure. It has the lowest population density in the country, and comprises almost all Amazon ecosystem area in Brazil. These features explain the low coverage of water distribution systems in large North cities, actually the lowest among the Brazilian regions. Overall, the other Brazilian regions have shown being sublinear as most of infrastructure variable analyzed worldwide, with an almost linear behavior in the Northeast region. In the opposite way, the number of high knowledge employees in the service sector is clearly superlinear for all Brazilian regions, following the superlinear characteristic of high knowlegde/technologies variables.

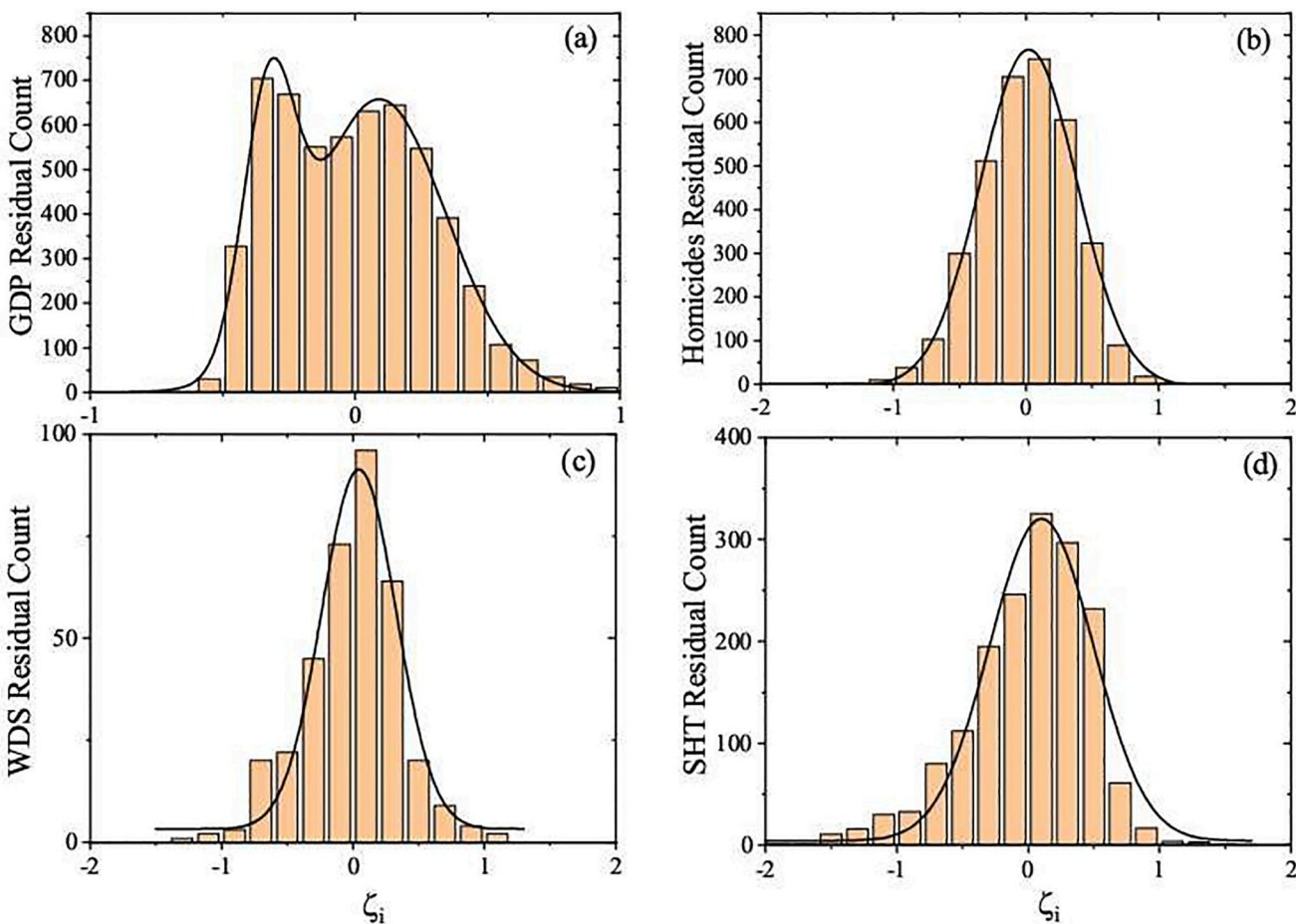

**Fig 6. Histograms of the residuals, $\zeta_i$, from the 2016 data: (a) GDP, (b) homicides, (c) water distribution system, and (d) the number of high knowledge employees in the service sector data of the ensemble Brazil.** The black lines are Gaussian curves ($\rho(\zeta_i) = A + B/(\omega\sqrt{\pi/2})\exp(-2(\zeta_i - \bar{\zeta}_i)^2/\omega^2)$ where $A$ and $B$ are constants, $\bar{\zeta}_i$ is the residual mean and $\omega$ the standard deviation). They have been used to describe the statistic of $\zeta_i$ for all but the GDP data, which is approximated by a bimodal Gaussian distribution. The corresponding histograms for all other years are very similar.

Fig 5(d) shows that the Northeast region has the largest proportional increase in the number of high knowledge employees in the service sector when one considers small, medium, and large cities, represented by the highest values of $\beta$ among all regions.

The analyzes based on the residual distribution makes it possible a relative comparison between the cities. The negative (positive) residual values mean that the cities are below (above) the average behavior for the same size range. Whether being above or below the average behavior means a good or bad scenario depends on the nature of the urban index. For the indexes studied in this paper, being above the average behavior is a good sign only for the GDP. However, this is not actually observed when we consider the ensemble of Brazilian cities, as we see in Fig 6(a), which shows that most Brazilian cities have GDP below the expected scaling value. When we consider the regional ensembles, most of the cities are near the average scaling behavior, as shown in Fig 7. We have investigated the UIR distribution over time for all sets. The shape of the UIR distribution can be well described by the Gaussian function. As a matter of fact, a bi-modal UIR distribution appears only for the GDP involving the complete Brazil set, as shown in Fig 6. Such distribution shows that most Brazilian cities stay below the average expected values predicted by the scaling analysis represented by Eq 4. Fig 6 shows the

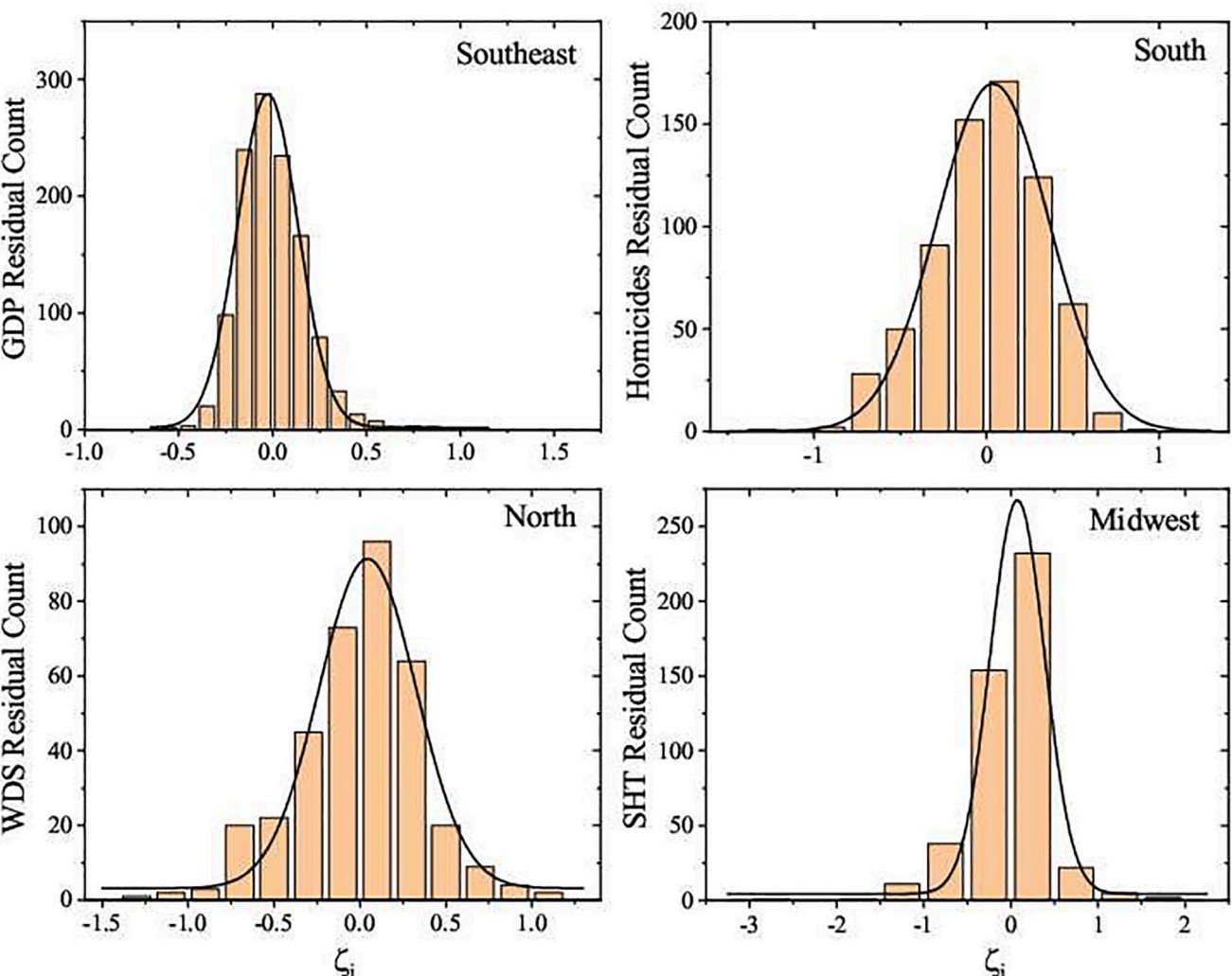

**Fig 7. Histograms of the residuals, $\zeta_i$, from the 2016 data: GDP, homicides, water distribution system and the number of high knowledge employees in the service sector for the Brazilian political regions.** The restricted sample of 4 out the 20 obtained curves is representative, once for all studied urban indicators and regions we have identified the same Gaussian shape. The black lines are Gaussian curves ($\rho(\zeta_i) = A + B/(\omega\sqrt{\pi/2}) \exp(-2(\zeta_i - \bar{\zeta}_i)^2/\omega^2)$ where $A$ and $B$ are constants, $\bar{\zeta}_i$ is the residual mean and $\omega$ the standard deviation), that describe the statistic of $\zeta_i$. The histograms for all other years are very similar.

UIR distributions for GDP, homicides, water distribution system and the number of high knowledge employees in the service sector in the complete Brazil set in 2016, while in Fig 7 we draw similar results for all regions and urban indexes. The UIR distribution graphs considering all indexes for every region over the years are not shown here because their shape are essentially the same.

Considering the Gaussian approximations to the UIR distributions, we have quantified how their asymmetry changed over time based on their skewness. Positive skewness values indicate right side asymmetry ($\zeta_i < 0$), i.e., larger number of cities below the average expected behavior. On the other hand, negative skewness (left side asymmetry) indicates that more cities are above the average expected behavior, $\zeta_i > 0$. As shown in Fig 8, skewness are positive for GDP over the years for all regions, being the largest values obtained for Northeast region. This indicates, therefore, that most cities are below the average scaling behavior. We also detected

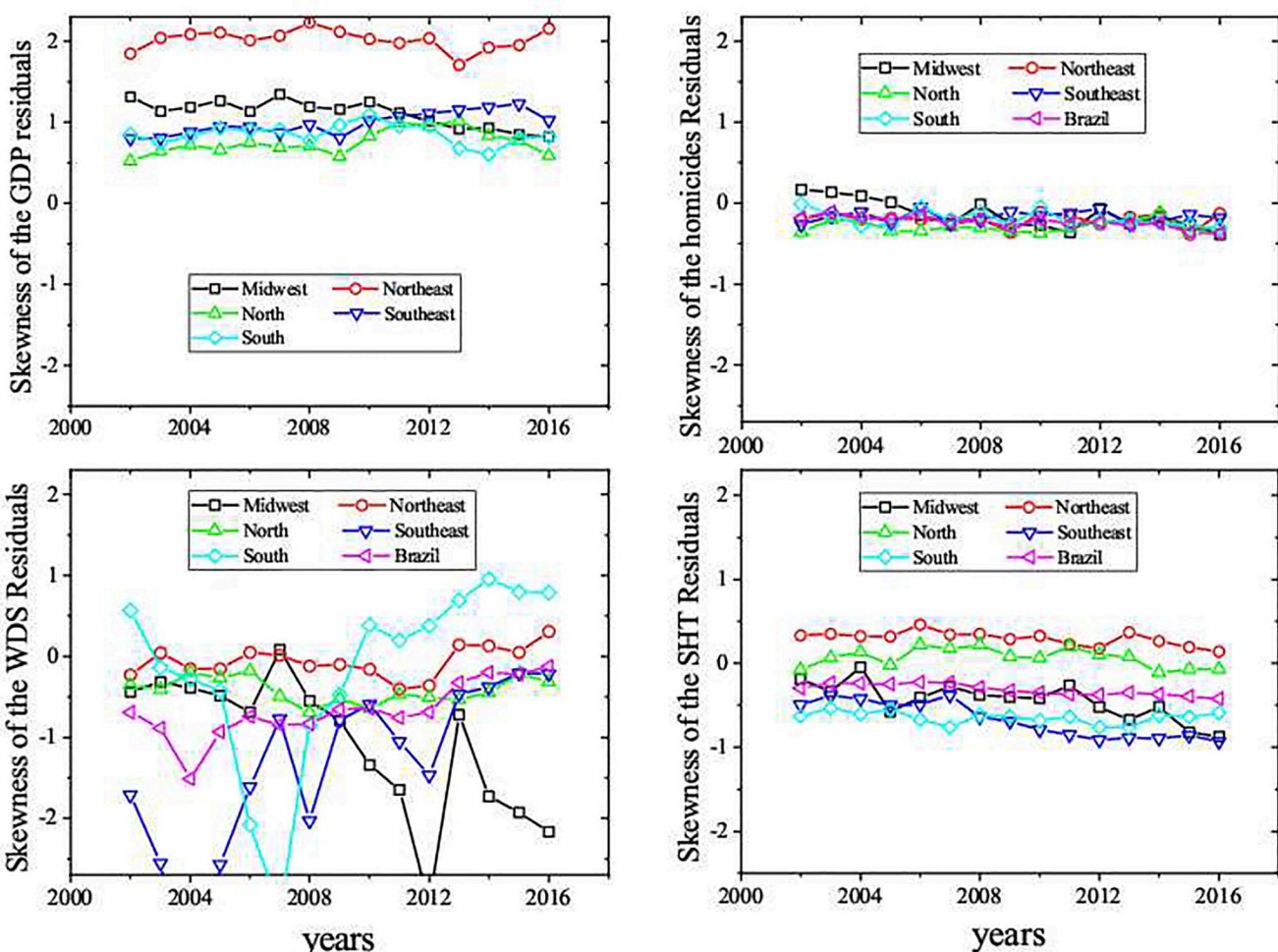

**Fig 8. Longitudinal analysis of the skewness of the residual distributions.** We have not done such analysis for the GDP of ensemble Brazil once its $\zeta$ distributions follow a bimodel Gaussian distribution.

an increasing trend of the Southeast region skewness, while the Midwest region has shown a opposite behavior. In the last decade, more small and medium cities in the Midwest region had a systematically increase in their GDP, becoming larger than the expected average value, while the Southeast has shown an opposite tendency. Again, the South and North regions presented similar behavior, showing a small right asymmetry.

The homicide data lead to UIR distributions with a slight left asymmetry for all ensembles, indicating that most cities are above the average expected scaling behavior. As the homicide is a collateral consequence of agglomeration, such characteristic is undesired, since being above the average means that in such cities the homicides cases even are higher than the average scaling prediction. For water distribution system, the skewness of the residuals of the Brazilian regions fluctuates around zero over time, which may be a good sign as it represents a natural random fluctuation. Some exceptions occur in the Midwest region for the last years. Regarding the the number of high knowledge employees in the service sector (SHT) residuals (see Fig 8), we can identify different behaviors: whereas the Midwest and Southeast regions became more left asymmetric, the South region kept the intensity of its left asymmetry. The North region

oscillates between right asymmetry and symmetric distributions, while the Northeast has preserved its right asymmetry over time.

Comparisons among city indicators, identifying critical states and cases of extreme conditions, e.g., large number of homicides, are usually used by policy makers in order to adopt well succeeded strategies to improve citizen living conditions and to plan future actions. Usually, such comparisons are made by per capita rates and number of case per 100M, both of which neglect the scaling dependency between the indicator and population size. This may lead to the adoption of wrong policies to undermine this effect.

Finally, to provide a deeper insight into the properties resulting from scaling analysis, we have produced and compared two ranks of all Brazilian cities based on the 2016 data for GDP, homicide, water distribution system and the number of high knowledge employees in the service sector. One of the ranks corresponds to the usual one, based on the number of cases per 100M inhabitants (or per capita for the GDP data), while the second rank was generated considering the highest residual values resulting from the scaling analysis. While the residual of each city is quantified by their distance to the average scaling behavior, which takes into account all Brazilian cities, the rate of cases per 100M just considers the linear dependency of the urban index and population size, independently of the characteristic of the ensemble. Fig 9 illustrates the results for both ranks. The yellow and blue dots indicate, respectively, the cities belonging the usual and the scaling analysis rank. The red dots represent the cities belonging to both ranks. It reveals a clear difference between the results produced by distinct criteria, for homicide and water distribution system data, which have shown, respectively, strong superlinear and sublinear scaling dependency. As expected, due to weak scaling dependency for GDP, both ranks comprise almost the same set of cities. Despite this fact, it is important to observe that, as a secondary effect, the position of the cities in the rank have changed for most of the cities. Moreover, the average population size in the rank generated by cases per 100M is considerably larger than that obtained from the scaling rank.

## Conclusions

Vast territorial countries are usually characterized by regions with different economic, social, infrastructure and cultural characteristics. Once the socioeconomic indicators reflect these non-homogeneities, it is natural to wonder whether they also influence the results of urban scaling analyzes. As a general rule, agglomeration effects have a strong influence also at the regional level. After performing a regional scaling analysis based on economic, violence and health indicators, we were able to verify that the Brazilian regions present some similar features in the scaling dependency of regional economic and health urban indicators. It is important to register the high quality of the Brazilian data from DATASUS and IBGE, specially when compared to other middle income countries, as India [27]. Furthermore, amazing findings of our analyzes reveal peculiarities among scaling properties across the different regions. Indeed, the comparison between results for the richest and poorest Brazilian regions show stronger scaling similarities than those resulting from the comparison between the results for the richest and second-richest regions. Also, comparable values of scaling exponent for regions with huge differences, as is the case of the South and North regions, appear rather unexpectedly. Indeed, despite comparable exponents, the South has a much larger population, is economically more developed and industrialized, and has a better infrastructure than the North region. Being a country with huge inequalities at many different levels, Brazil provides an ideal case study for the observation of regional scaling properties, which allow for such surprising results. Our results suggest that the same kind of regional approach presented here, besides being directly applicable to countries with similar features, might also be well adapted for smaller and more

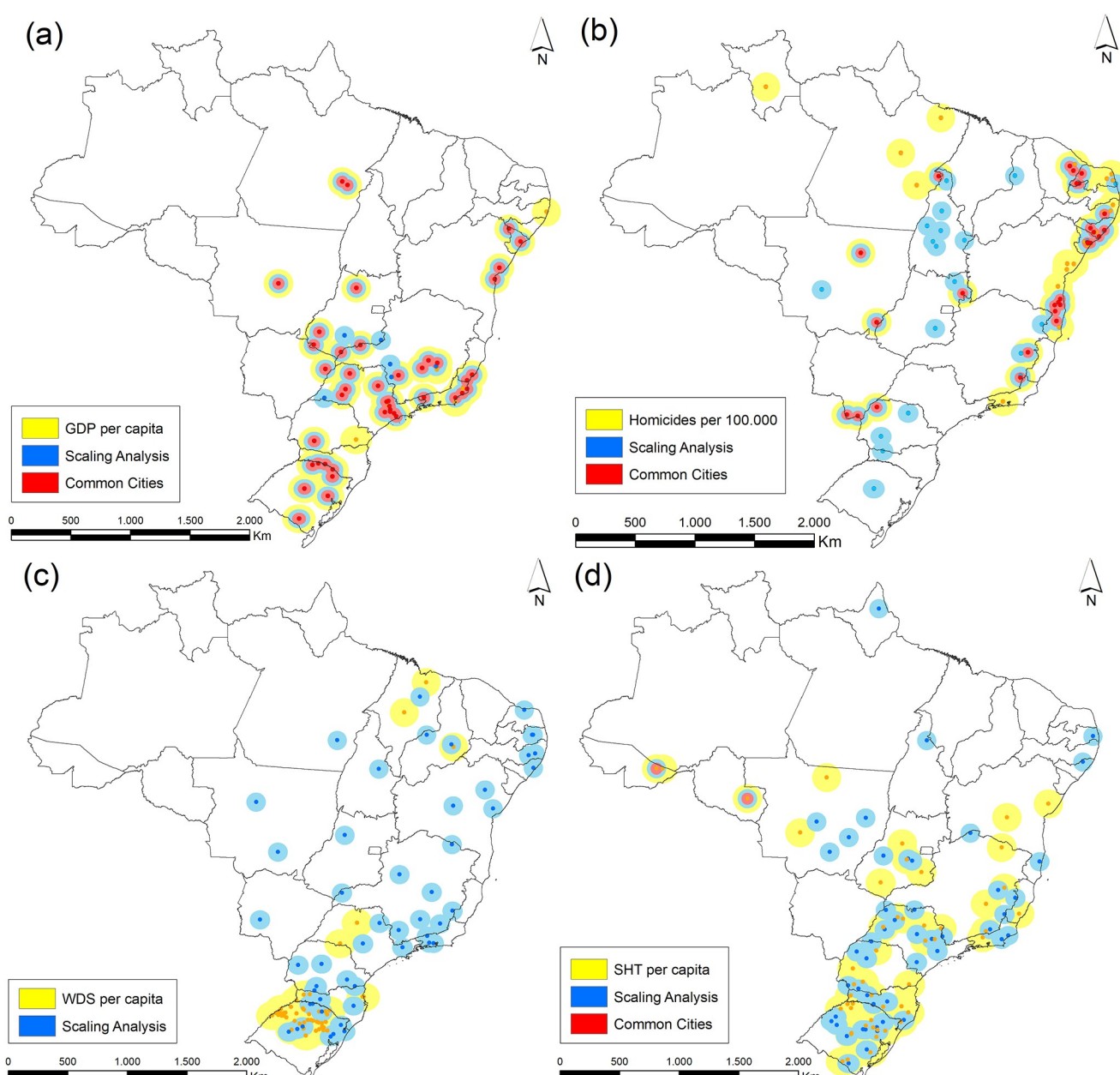

**Fig 9. Maps showing the location of the top fifty ranked Brazilian cities according to the following criteria: (a) Brazilian GDP, (b) most violent Brazilian cities according to the number of homicides, (c) length of the water distribution system, and (d) the number of high knowledge employees in the service sector, in 2016.** The yellow dots indicate the cities according to the cases per 100.000 rank, while the blue dots represent the cities of the scaling rank generated by the residuals of the cities. The red dots represent the cities that appeared in both ranks. Maps generated with the free and open source geographic information system (QGIS) version 3.4 LTR.

homogeneous cases. It worth to point out that the definition of a city is still a discussion topic and depending on the focus of the investigation different interpretations may arise [35, 36]. Also, the reasons why most of the urban indicators have followed the power law dependency is still a research question [37].

We have emphasized that the use of per capita urban indicators overlooks the presence of the non-linear scaling relations for the four analyzed cases, a fact that may also happen to

other urban features affected by non-linear interactions produced by the agglomeration phenomena. Indeed, the rank produced based on the residuals with respect to the scaling analyzes has sensible differences to that based on per capita indicators. We understand that a regional based scaling analysis may help scholars and policymakers get a more comprehensive view of urbanization related problems, being able to interfere locally in a more specific way rather than by uniform public actions designed for the whole country. Our analysis intend to initiate a discussion about the influence of the regional Brazilian heterogeneities over the scaling dependency of the urban variables. This may be evidenced by the differences in the scaling dependency of the GDP of the richest Brazilian regions, Southeast and South. Also, the Northeast and North regions despite of being the poorest Brazilian regions, appearing with considerable differences on the scaling dependency of the high knowledge employees. Therefore, as also verified by Bettencourt from his study about India [27], such regional analysis may show which regions of continental countries should have priority to social-economic policies implementations in order to develop, in a efficient way, all country. Frequently the Brazilian govern do not consider the hererogeneities. An example is the new cash transfer programme due to the SARS-CoV2 (COVID-19) pandemic. The main criteria to be eligible to receive the money was being unemployed. Once one is eligible, the govern just stipulated a fix value that a person may receive. The problem of such general programme is that people who live in capitals and in developed regions as South and Southeast are less assisted than those living in other regions mainly due to the cost of life. Despite we have limited our analysis to 4 urban variables, we have selected them in such a way that they cover distinct and representative urban characteristics as economics, violence and health, infrastructure and education. Thus, they are able to provide a general view confirming the existence of important differences and similarities among the urban scaling properties of socio-economic indicators from distinct regions in a same country.

## Supporting information

**S1 Appendix.**
(DOCX)

## Author Contributions

**Conceptualization:** Caio Porto de Castro, Gervásio Ferreira dos Santos, Roberto Fernandes Silva Andrade, Maurício Lima Barreto.

**Formal analysis:** Caio Porto de Castro.

**Investigation:** Caio Porto de Castro.

**Methodology:** Caio Porto de Castro, Gervásio Ferreira dos Santos, Roberto Fernandes Silva Andrade.

**Software:** Caio Porto de Castro.

**Visualization:** Caio Porto de Castro, Anderson Dias de Freitas.

**Writing – original draft:** Caio Porto de Castro, Gervásio Ferreira dos Santos, Roberto Fernandes Silva Andrade, Maurício Lima Barreto.

**Writing – review & editing:** Caio Porto de Castro, Gervásio Ferreira dos Santos, Maria Isabel dos Santos, Roberto Fernandes Silva Andrade, Maurício Lima Barreto.

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
