## [Decision Letter · Decision Letter 0]

17 Apr 2020

PONE-D-20-02779

Socio-economic urban scaling properties: influence of regional geographic heterogeneities in Brazil

PLOS ONE

Dear Dr. de Castro,

Thank you for submitting your manuscript to PLOS ONE. After careful consideration, we feel that it has merit but does not fully meet PLOS ONE’s publication criteria as it currently stands. Therefore, we invite you to submit a revised version of the manuscript that addresses the points raised during the review process.

We would appreciate receiving your revised manuscript by Jun 01 2020 11:59PM. To enhance the reproducibility of your results, we recommend that if applicable you deposit your laboratory protocols in protocols.io, where a protocol can be assigned its own identifier (DOI) such that it can be cited independently in the future. For instructions see: http://journals.plos.org/plosone/s/submission-guidelines#loc-laboratory-protocols

We look forward to receiving your revised manuscript.

Kind regards,

Mingxing Chen, Ph.D.

Academic Editor

PLOS ONE

2. We note that Figures 3 and 9 in your submission contain map images which may be copyrighted. All PLOS content is published under the Creative Commons Attribution License (CC BY 4.0), which means that the manuscript, images, and Supporting Information files will be freely available online, and any third party is permitted to access, download, copy, distribute, and use these materials in any way, even commercially, with proper attribution. For these reasons, we cannot publish previously copyrighted maps or satellite images created using proprietary data, such as Google software (Google Maps, Street View, and Earth). For more information, see our copyright guidelines: http://journals.plos.org/plosone/s/licenses-and-copyright.

1.    You may seek permission from the original copyright holder of Figures 3 and 9 to publish the content specifically under the CC BY 4.0 license. 

3. We note that in your methods section you provide the reference citation for the dataset used in the study. We would like to recommend that you also include a link to the data in the methods section such that readers can more easily access it.

Reviewers' comments:

Reviewer's Responses to Questions

**Comments to the Author**

1. Is the manuscript technically sound, and do the data support the conclusions?

Reviewer #1: Partly

Reviewer #2: Yes

2. Has the statistical analysis been performed appropriately and rigorously? 

Reviewer #1: Yes

Reviewer #2: Yes

3. Have the authors made all data underlying the findings in their manuscript fully available?

Reviewer #1: No

Reviewer #2: Yes

4. Is the manuscript presented in an intelligible fashion and written in standard English?

Reviewer #1: Yes

Reviewer #2: Yes

5. Review Comments to the Author

Reviewer #1: Thank you for the opportunity to review this interesting paper. While I found it to be reasonably well-written and interesting, I cannot recommend publication at this time. After reading the manuscript carefully, I think the scientific contribution and novel knowledge of the research need to be specified. What problems should be solved, such as theory, method or policy of urban development in Brazil? Compared with the relevant research, where is the innovation and contribution of this study? I think some clarifications and major alterations will increment the general quality of the paper.

1.Authors need to better address the goals of the paper in the end of the Introduction section. They are poorly or insufficiently addressed.

2.Data sources require explanation and critical discussion.

3.Figure 3 and Figure 9 could be improved by drawing.

4.Line 62: the title "Discussion" is not appropriate.

5.Further analysis on the influence of regional geographic heterogeneities is important.

Reviewer #2: Using economic, infrastructure and violence related data sets for the time interval 2002-2016 in Brazil, the study analyzed socio-economic urban scaling properties in the country. Results indicate that regional specificities related to some selected factors have a larger influence on the absolute value of the urban indexes. Regional scaling similarities and differences among regions were also uncovered. The study was good writing, and these findings are helpful for understanding the socio-economic urban scaling properties. However, there are some issues the authors must response before this manuscript to be published.

1, the study pointed that there exists regional heterogeneities in Brazil, the reviewer suggest authors test the statistical significant of the spatial stratified heterogeneities using statistic methods.

2, comparison of findings between the study and the other related papers should be discussed in the manuscript.

3, limitations and the further studies of the study should be discussed in the manuscript.

4, some findings have been discussed, however some of them need more scientific interpretations.

5, the reviewer suggest authors check structure of the manuscript. For example, in line 62, “discussion” is appeared before “methods” section.

6. PLOS authors have the option to publish the peer review history of their article (what does this mean?). If published, this will include your full peer review and any attached files.

Reviewer #1: No

Reviewer #2: No

---

## [Author Response · Author response to Decision Letter 0]

23 Jun 2020

Reviewer #1: Thank you for the opportunity to review this interesting paper. While I found it to be reasonably well-written and interesting, I cannot recommend publication at this time. After reading the manuscript carefully, I think the scientific contribution and novel knowledge of the research need to be specified. What problems should be solved, such as theory, method or policy of urban development in Brazil? Compared with the relevant research, where is the innovation and contribution of this study? I think some clarifications and major alterations will increment the general quality of the paper.

Author’s response: The authors thank the reviewer for reading our manuscript and for the important comments in the report. Overall, we believe that after the changes that we introduced in the new version, we have a significant improvement on our manuscript.

1.Authors need to better address the goals of the paper in the end of the Introduction section. They are poorly or insufficiently addressed.

1. As opportunely suggested by the reviewer we have included a new paragraph in the end of the introduction where we clearly address the goals of our work. Please, see text from line 62 to line 76.

2.Data sources require explanation and critical discussion.

2. We have included an additional explanation on the data in the first paragraph of the section “Results and Discussion”. Please, see text from line 124 to line 130.

3.Figure 3 and Figure 9 could be improved by drawing.

3. We have amplified Figure 9 and improved Figure 3. We hope that after the changes the figures have improved their quality.

4.Line 62: the title "Discussion" is not appropriate.

4. We thank the referee for calling our attention to this issue. We have removed “Discussion”, in line 62.

5.Further analysis on the influence of regional geographic heterogeneities is important.

5. We do agree that further analysis in order to understand completely how socio-economic heterogeneities present in continental size countries. This work intends to contribute to such discussion in Brazil, and certainly may be extended to include a larger number of urban variables. For the moment, we understand our work already uncovers a number of interesting findings. Indeed, the quantities selected for our analysis are representative of several sectors of socio-economic indicators, providing a first but wide view of their scaling properties. We intend to proceed further with our investigation, so that new insights coming from our results may be presented in a new future. 

Reviewer #2: Using economic, infrastructure and violence related data sets for the time interval 2002-2016 in Brazil, the study analyzed socio-economic urban scaling properties in the country. Results indicate that regional specificities related to some selected factors have a larger influence on the absolute value of the urban indexes. Regional scaling similarities and differences among regions were also uncovered. The study was good writing, and these findings are helpful for understanding the socio-economic urban scaling properties. However, there are some issues the authors must response before this manuscript to be published.

Author’s response: The authors thank the reviewer for reading our manuscript and for the important comments in the report. Overall, we believe that after the changes that we introduced in the new version, we have a significant improvement on our manuscript.

1, the study pointed that there exists regional heterogeneities in Brazil, the reviewer suggest authors test the statistical significant of the spatial stratified heterogeneities using statistic methods.

1. We have added a supporting information material which presents the Brazilian spatial heterogeneities and also the regional ones.

2, comparison of findings between the study and the other related papers should be discussed in the manuscript.

2. In the section “Conclusion” we have mentioned that our conclusions are in the same direction of the Bettencourt paper, ref. 27, which presents regional scaling analysis for India.

3, limitations and the further studies of the study should be discussed in the manuscript.

3. We have include a new paragraph in the section “conclusion”.

4, some findings have been discussed, however some of them need more scientific interpretations.

4. We believe that we have improve the discussion of our findings by including a new paragraphs in the “Introduction” and in the “Conclusion” sections.

5, the reviewer suggest authors check structure of the manuscript. For example, in line 62, “discussion” is appeared before “methods” section.

5. We thank the referee for calling our attention to this issue. We have removed “Discussion”, in line 62.

---

## [Decision Letter · Decision Letter 1]

11 Aug 2020

PONE-D-20-02779R1

Socio-economic urban scaling properties: influence of regional geographic heterogeneities in Brazil

PLOS ONE

Dear Dr. de Castro,

Thank you for submitting your manuscript to PLOS ONE. After careful consideration, we feel that it has merit but does not fully meet PLOS ONE’s publication criteria as it currently stands. Therefore, we invite you to submit a revised version of the manuscript that addresses the points raised during the review process.

I have received mixed feedback from the two reviewers. While Reviewer 1 is satisfied with your revision and reply, Reviewer 2 suggests another round of major revision before this manuscript can move forward. 

We look forward to receiving your revised manuscript.

Kind regards,

Mingxing Chen, Ph.D.

Academic Editor

PLOS ONE

Reviewers' comments:

Reviewer's Responses to Questions

**Comments to the Author**

1. If the authors have adequately addressed your comments raised in a previous round of review and you feel that this manuscript is now acceptable for publication, you may indicate that here to bypass the “Comments to the Author” section, enter your conflict of interest statement in the “Confidential to Editor” section, and submit your "Accept" recommendation.

Reviewer #1: All comments have been addressed

Reviewer #2: All comments have been addressed

2. Is the manuscript technically sound, and do the data support the conclusions?

Reviewer #1: Yes

Reviewer #2: Partly

3. Has the statistical analysis been performed appropriately and rigorously? 

Reviewer #1: Yes

Reviewer #2: Yes

4. Have the authors made all data underlying the findings in their manuscript fully available?

Reviewer #1: Yes

Reviewer #2: Yes

5. Is the manuscript presented in an intelligible fashion and written in standard English?

Reviewer #1: Yes

Reviewer #2: Yes

6. Review Comments to the Author

Reviewer #1: All comments have been addressed. The present document has been largely improved from its previous version.

Reviewer #2: This version has been improved much. Some issues, however, should be addressed. Regional geographic heterogeneities of the research object is the core content of the article, in the manuscript, the authors test the statistical significant of the spatial stratified heterogeneities using Local Spatial Autocorrelation Moran Index (LISA). The reviewer think that LISA can be used to test the local heterogeneities, however, the regional heterogeneity studied in the manuscript is more suitable to be expressed by spatial hierarchical heterogeneity, which can be calculated by geodetector q statistic (Wang, et al. A Measure of Spatial Stratified Heterogeneity. Ecological Indicators 2016), so that the reviewer suggest authors complement the analysis and added in the main text.

7. PLOS authors have the option to publish the peer review history of their article (what does this mean?). If published, this will include your full peer review and any attached files.

Reviewer #1: No

Reviewer #2: No

---

## [Author Response · Author response to Decision Letter 1]

28 Sep 2020

Response to the Reviewers

We would like to thank the editor and reviewers for their contributions. Please, find below, in red color, our point by point comments to the reviewers.

Reviewer #1: All comments have been addressed. The present document has been largely improved from its previous version.

Author’s response: The authors thank the reviewer for the previous comments. We believe that after the changes, we have an improvement in our manuscript.

Reviewer #2: This version has been improved much. Some issues, however, should be addressed. Regional geographic heterogeneities of the research object is the core content of the article, in the manuscript, the authors test the statistical significant of the spatial stratified heterogeneities using Local Spatial Autocorrelation Moran Index (LISA). The reviewer think that LISA can be used to test the local heterogeneities, however, the regional heterogeneity studied in the manuscript is more suitable to be expressed by spatial hierarchical heterogeneity, which can be calculated by geodetector q statistic (Wang, et al. A Measure of Spatial Stratified Heterogeneity. Ecological Indicators 2016), so that the reviewer suggest authors complement the analysis and added in the main text.

Author’s response: The authors thank the reviewer for this new suggestion and fully agree with it. We have included new results and a discussion of the parameter q, reported in Wang, et al. A Measure of Spatial Stratified Heterogeneity, Ecological Indicators 2016. In fact, the results we have found support our discussion regarding regional geographic heterogeneities. Metropolitan areas in every Brazilian region concentrates wealth, infrastructure, high level education employees and violence, which explain the values of q close to zero (heterogeneity within the regions are more significant than between them). The regions present similar heterogeneity, despite the discrepancy on the absolute value of the socioeconomic variables. On the other hand, the largest value of q associated with WDS indicates a comparatively smaller infrastructure discrepancy between the richest regions with the poorest ones. This can be explained, among other reasons as political and historical, by the fact that the poorer regions, as North and Northeast, have globally had less investments in infrastructure. We have included such discussion (in red) in pages 5 and 6 of the manuscript.

---

## [Decision Letter · Decision Letter 2]

21 Oct 2020

PONE-D-20-02779R2

Socio-economic urban scaling properties: influence of regional geographic heterogeneities in Brazil

PLOS ONE

Dear Dr. de Castro,

Thank you for submitting your manuscript to PLOS ONE. After careful consideration, we feel that it has merit but does not fully meet PLOS ONE’s publication criteria as it currently stands. Therefore, we invite you to submit a revised version of the manuscript that addresses the points raised during the review process.

I am glad to inform you that after the second round of reviews, the paper was accepted with minor revisions. 

We look forward to receiving your revised manuscript.

Kind regards,

Mingxing Chen, Ph.D.

Academic Editor

PLOS ONE

Reviewers' comments:

Reviewer's Responses to Questions

**Comments to the Author**

1. If the authors have adequately addressed your comments raised in a previous round of review and you feel that this manuscript is now acceptable for publication, you may indicate that here to bypass the “Comments to the Author” section, enter your conflict of interest statement in the “Confidential to Editor” section, and submit your "Accept" recommendation.

Reviewer #2: All comments have been addressed

2. Is the manuscript technically sound, and do the data support the conclusions?

Reviewer #2: Yes

3. Has the statistical analysis been performed appropriately and rigorously? 

Reviewer #2: Yes

4. Have the authors made all data underlying the findings in their manuscript fully available?

Reviewer #2: Yes

5. Is the manuscript presented in an intelligible fashion and written in standard English?

Reviewer #2: Yes

6. Review Comments to the Author

Reviewer #2: The present document has been largely improved from its previous version.

In addition, in table1, it’s better to replace the statistical symbol q with Geodetector q.

7. PLOS authors have the option to publish the peer review history of their article (what does this mean?). If published, this will include your full peer review and any attached files.

Reviewer #2: No

---

## [Author Response · Author response to Decision Letter 2]

28 Oct 2020

Reviewer #2: The present document has been largely improved from its previous version.

In addition, in table1, it’s better to replace the statistical symbol q with Geodetector q.

Author’s response: The authors thank the reviewer for this suggestion. We have replaced the symbol q with Geodetector q in table 1.

---

## [Editor Report · Decision Letter 3]

10 Nov 2020

Socio-economic urban scaling properties: influence of regional geographic heterogeneities in Brazil

PONE-D-20-02779R3

Dear Dr. de Castro,

We’re pleased to inform you that your manuscript has been judged scientifically suitable for publication and will be formally accepted for publication once it meets all outstanding technical requirements.

Kind regards,

Mingxing Chen, Ph.D.

Academic Editor

PLOS ONE
---

## [Editor Report · Acceptance letter]

17 Nov 2020

PONE-D-20-02779R3 

Socio-economic urban scaling properties: influence of regional geographic heterogeneities in Brazil  

Dear Dr. de Castro:

I'm pleased to inform you that your manuscript has been deemed suitable for publication in PLOS ONE. Congratulations! Your manuscript is now with our production department. 

Kind regards, 

on behalf of

Prof. Mingxing Chen 

Academic Editor

PLOS ONE